# Do Magnetic murmurs guide birds? A directional statistical investigation for influence of Earth's Magnetic field on bird navigation

**Prithwish Ghosh** [1] *, **Debashis Chatterjee**[2], **Amlan Banerjee**[3], **Shiladri Shekhar Das**[3]

**1** Department of Statistics, North Carolina State University, Raleigh, NC, United States of America,
**2** Department of Statistics, Siksha Bhavana (Institution of Science), Visva Bharati, Bolpur, Santiniketan, India,
**3** Geological Studies Unit, Indian Statistical Institute, Kolkata, West Bengal, India

* pghosh4@ncsu.edu

**Data Availability Statement:** The dataset is uploaded in the Harvard Data verse link: https://doi.org/10.7910/DVN/MWDULG.

## Abstract

This paper delves into the intricate relationship between changes in Magnetic inclination and declination at specific geographical locations and the navigational decisions of migratory birds. Leveraging a dataset sourced from a prominent bird path tracking web resource, encompassing six distinct bird species' migratory trajectories, latitudes, longitudes, and observation timestamps, we meticulously analyzed the interplay between these avian movements and corresponding alterations in Magnetic inclination and declination. Employing a circular von Mises distribution assumption for the latitude and longitude distributions within each subdivision, we introduced a pioneering circular-circular regression model, accounting for von Mises error, to scrutinize our hypothesis. Our findings, predominantly supported by hypothesis tests conducted through circular-circular regression analysis, underscore the profound influence of Magnetic inclination and declination shifts on the dynamic adjustments observed in bird migration paths. Moreover, our meticulous examination revealed a consistent adherence to von Mises distribution across all bird directions. Notably, we unearthed compelling correlations between specific bird species, such as the Black Crowned Night Heron and Brown Pelican, exhibiting a noteworthy negative correlation with Magnetic inclination and a contrasting positive correlation with Magnetic declination. Similarly, the Pacific loon demonstrated a distinct negative correlation with Magnetic inclination and a positive association with Magnetic declination. Conversely, other avian counterparts showcased positive correlations with both Magnetic declination and inclination, further elucidating the nuanced dynamics between avian navigation and the Earth's magnetic field parameters.

## Introduction

Migration describes the recurring annual movements, particularly observable for certain species of birds known as migratory birds. Many migratory birds make incredible journeys each

**Funding:** The author(s) received no specific funding for this work.

**Competing interests:** The authors have declared that no competing interests exist.

year. Mobile connectivity is essential because birds can be affected at any part of their life cycle and location [1]. The migration orientation of birds and other animals has been interesting in the literature for decades. For instance [2, 3], addressed the orientation processes of migrating birds concerning sunset and sunrise. addressed the migration orientations of tree sparrows. The ability to detect the direction of ambient magnetic fields provides an important directional signal to birds. It may be the basis for the navigational skills of many different species [4]. The initial confirmation of a magnetic compass in birds was established through research on European Robins, *erubecula* [5, 6], achieved by observing their migratory orientation behavior during migration periods. Additionally, they exhibited a response to alterations in magnetic North by adjusting their headings accordingly [6, 7], showing that caged migratory garden warblers perform head-scanning behavior well suited to detect this magnetic symmetry plane. Growing up in an altered magnetic field may affect the initial orientation of young homing pigeons [8]. In the natural geomagnetic field, birds move toward their migratory direction after head scanning [9]. The researchers could analyze the functional properties of the compass mechanism in birds by conducting experiments focused on migratory orientation in robins and conditioning experiments involving chickens. These investigations unveiled two unexpected characteristics of this navigational system [10]. The orientations of the circular distributions of the escape attempts of birds or the vanishing points in free flight have been subjected to refined statistical analysis, [11, 12]. Review literature like [6] exists on the directional orientation of birds with the help of geomagnetic fields under various light conditions. On the contrary, very little literature addresses the mathematical models for directional change patterns of migratory bird paths on the Earth concerning relevant geological parameters using Directional and Spherical Statistics [13, 14]. Use of circular statistical tools for bird migration-related queries. [15] analyzed bird directional data using instrumental and spatial statistical methods. Their work focused on bar-tailed godwits tracked via satellite telemetry during their remarkable non-stop flights across the Pacific Ocean (also documented in [16]).

[17] explored the potential role of magnetoreception in avian navigation and investigated clusters of superparamagnetic magnetite particles found in the upper-beak skin of homing pigeons, suggesting a biological basis for magnetic field sensitivity.

[18] presented a pioneering work analyzing avian migration patterns. They emphasized the importance of rigorous quantitative data analysis and mathematical modeling to gain a deeper understanding of the dynamic processes underlying bird migration ([18]). Similarly, [19] highlighted the limitations of the traditional "clock-and-compass" model in explaining bird distribution patterns. They stressed the need for further research incorporating uncertainties, landmass distributions, and additional factors influencing avian navigation strategies ([19]).

The rest of the paper is as follows. Section essentially includes the directional statistical preliminaries used in this research work, details about obtaining the Angular values of Magnetic field change on the Earth, Haversine methods to get the birds' direction change, proposed novel circular-circular regression model applicable in this context. Section contains the proposed circular-circular regression model, its goodness of fit, and parameter estimations. Section includes a brief theory for testing for directional correlation applied in this context. Section provides for the development of our analysis. Section discusses our methods and result interpretation. We conclude in section. In the Fig 1 we have plotted the bird dataset all over the world map with respect to the species. As per the existing literture we get that Swainson's hawks breed in the grasslands of the Midwestern and Western United States, commonly constructing nests within small clusters of trees adjacent to streams and agricultural fields, Pacific loons primarily inhabit the open waters of the ocean, except for a three-month breeding season during which they reside on freshwater lakes in northern Canada, Alaska, and Siberia.

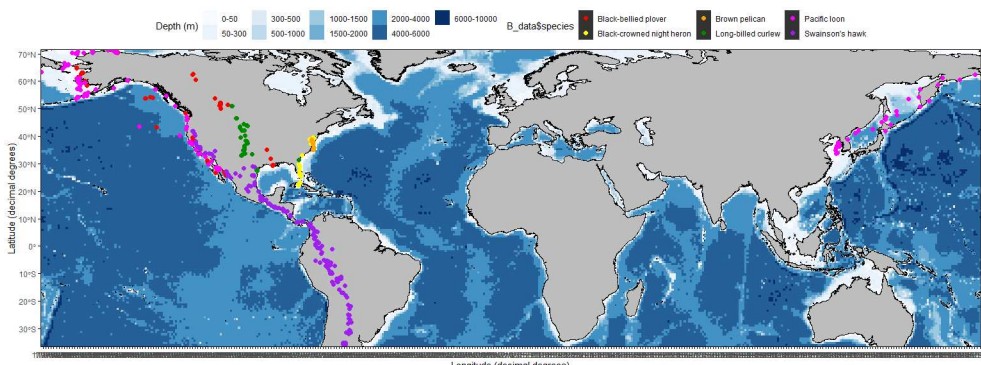

**Fig 1. The observed birds for all species(mentioned with different colors) are plotted over the world map, located in the American sub-continent, and some in Japanese continents(The plots are generated by using R and ggOceanmap Package).**

During the summer breeding season, Long-billed curlews inhabit prairies, pastures, and agricultural fields across western North America, The brown pelican, a distinctive and sizable bird, thrives in colonies along shorelines and small islands, often numbering in the thousands. This species is prevalent along the coastlines of the southern and western United States, as well as in the Caribbean, Mexico, Central America, and northern South America, Throughout the breeding season, black-crowned night herons inhabit various regions across North America, with notably larger populations congregating along the coasts, During the summer breeding seasons, North American black-bellied plovers inhabit the Arctic tundra of Canada and Alaska.

## Objective of the paper

The main objective of this paper can be summarized as follows. We intend to use a circular-circular regression model to check the birds' directional change dependency with independent covariates. Statistically speaking, the dependent variable is the Directional change of birds (we explained in section how we defined and computed it starting from the raw dataset), and the independent variables are the positional change of the Magnetic field (we explained in section the definition and calculated it starting from the raw dataset). Compared to the previous works, the novelty of this paper lies in taking a proper directional statistical approach to unveil directional correlation among variables of our interest.

In this paper, we take a novel directional statistical approach to assert whether the direction of the migratory bird paths is affected by Magnetic field change. For that, we propose a novel hypothesis that the change in the direction of the migratory bird paths is affected by Magnetic field change and conduct a novel directional statistical model-based hypothesis testing based on that. We start from the assumption that errors follow Von Mises distributions. We discuss the details of compiling data set 2, which, in addition to dataset 1, contains two additional columns: column $\theta$ containing the angle values of the direction change of the path between two consecutive path points of a given species. We use the Haversine formula to get the birds' direction change. Observe that the measurements considered in this paper are angular directions. The circular and spherical representation of the directions represented by angles falls under the paradigm of a unique statistical field called directional statistics. Naturally, the use of directional statistical tools is justified. Although scientists once regarded it as a proposition, the influence of natural magnetic fields on bird migratory behavior is more or less accepted

now. Regardless, literature addressing the rigorous directional statistical assertion that a bird's decision to change its migration path statistically significantly correlates with the change of the Earth's magnetic field is inadequate. In this paper, we intend to fill that gap.

While there's existing research on bird migration, to our finding, a substantial gap exists in analyzing their directional distribution patterns using directional statistics. Understanding how birds navigate their incredible migratory journeys is a complex puzzle. Traditional statistical tools struggle to analyze this phenomenon because bird migration data isn't simply about distance traveled; it's all about direction. Latitude and longitude data, the cornerstones of geographic location, paint a picture on a flat map, but migratory paths traverse a curved globe. Here's where directional statistical tools become crucial. They account for the Earth's spherical nature, allowing us to accurately analyze the angles and directions birds travel. Without these specialized tools, we risk misinterpreting the subtle influences of natural parameters like the Earth's magnetic field on bird navigation. Traditional methods often treat migration paths as simple point-to-point journeys, neglecting the crucial role of specialized statistical tools for handling directionality coming naturally from latitude and longitude. This oversight can lead to misinterpretations, particularly when investigating the influence of natural phenomena like the Earth's magnetic field on bird navigation.

Although prior studies (e.g., [15, 17–19]) employed various statistical methods to analyze bird navigation and migration patterns, they did not necessarily utilize directional statistics. However, their emphasis on rigorous quantitative analysis, the complexities of avian navigation strategies, and the limitations of traditional models strongly motivate the application of specialized statistical tools, such as circular statistics employed in this study, to achieve a more comprehensive understanding of this intricate phenomenon. By incorporating directionality into the analysis, we can unlock a deeper understanding of this remarkable feat of nature, avoiding potential blunders and revealing the secrets whispered by the "magnetic murmurs" that guide these feathered voyagers.

We analyze bird migratory paths using a dataset containing latitude, longitude, and observation dates. Subsequently, we construct a derived dataset incorporating both the directional changes in bird paths and the spatiotemporal variations in the Earth's magnetic field. Through circular-circular regression analysis and hypothesis testing on directional correlations between bird direction changes and environmental factors such as magnetic inclination and declination, we observe a statistically significant non-zero correlation between bird directional changes and the examined environmental factors. The novelty of our approach lies in the utilization of circular-circular regression, a specialized form of regression analysis tailored for circular data, to investigate the relationship between changes in bird path direction and fluctuations in the magnetic field. This departure from traditional linear regression methods enables us to effectively model and analyze the intricate interactions between bird migration patterns and variations in the magnetic field, thereby offering deeper insights into the navigational mechanisms of avian species.

## Materials and methods

Our methodologies involve a step-by-step process outlined as in the flow chart. First, we collect the necessary data (Step 1). Next, the data is partitioned based on species and sorted by location (Step 2). Following this, we calculate the bird's direction and the magnetic inclination and declination (Steps 3 and 4). Subsequently, we analyze the distribution of the change in bird direction using the Watson test (Step 5) and apply correlation tests between bird direction and magnetic field parameters (Step 6). Finally, circular regression analysis explores the relationship between bird direction and magnetic inclination and declination (Step 7). This sequential

approach allows for a systematic investigation into the interplay between bird navigation and magnetic field characteristics.

**Algorithm 1**: Bird Migration Analysis Workflow

**Data Collection & Processing**

1. Collect data on Bird movements (latitude, longitude, timestamps).
2. Gather/Build data on Earth's magnetic field. (we use formulas, for employing the world magnetic model [20]).
3. Derive dataset incorporating directional changes and spatiotemporal positions. (computed utilizing the Haversine method [21]).

**Hypothesis & Directional Statistical Analysis**

1. Propose novel hypothesis: changes in Earth's magnetic field influence bird migratory path decisions.
2. Apply the von Mises mixture model to raw and derived datasets and test goodness-of-fit.
3. Examine directional associations using circular-circular regression and hypothesis testing.
4. Assess correlations between bird path changes and environmental variables (magnetic).

**Interpretation & Hypothesis**

1. Introduce a spherical mixture model for further analysis and obtain optimal partitioning based on BIC (Bayesian Information Criteria).
2. Identify statistically significant correlations and their interpretation.

**Conclusion**

1. Gain insights into bird navigation and environmental factor influences.

## Novel hypothesis on decision of bird to change its migratory path based on the Chane of Earth Magnetic field

We seek a rigorous statistical answer to whether a suitable change of the Earth's Magnetic field at a spot on the Earth directly influences a bird's decision to fly on that spot to change its paths, and our statistical result affirms that.

A substantial body of research has explored the connection between bird navigation and the Earth's magnetic field (e.g., [7, 9, 15, 17–19]). Observations in natural geomagnetic fields demonstrate that birds align their movements with their migratory direction after head scanning [8]. Collectively, these findings support the hypothesis that bird migration is dependent on the Earth's magnetic fields. These studies provide a strong foundation for investigating the influence of magnetic fields on bird migration patterns. However, they often rely on traditional statistical methods that may not fully capture the directional nature of this phenomenon.

This study proposes a novel approach by recasting the existing hypothesis within a directional statistical framework (Bird migration patterns are influenced by variations in the Earth's magnetic field). This framework acknowledges that migratory paths and magnetic field variations are inherently directional data, best analyzed using specialized circular statistics.

We consider a novel circular-circular regression model for that and test whether the regression model is validated (discussed in section).

## World magnetic model

The World Magnetic Model (WMM) provides a comprehensive representation of the Earth's magnetic field on a global scale [22]. This model utilizes a spherical harmonic expansion up to the $12^{12}$ degree and order, capturing the magnetic scalar potential generated by the Earth's core, which contributes to the geomagnetic main field. Alongside the 168 spherical-harmonic

"Gauss" coefficients, the model incorporates an equal number of spherical-harmonic Secular-Variation (S.V.) coefficients. All quantities in this section adhere to the following measurement conventions: angles are in radians, lengths are in meters, magnetic intensities are in nanoteslas (nT, where one tesla is one weber per square meter or one $kg.s^{-2}.A^{-1}$), and times are in years [23].

The primary magnetic field $B_m$ is a potential field and can, therefore, be expressed in geocentric spherical coordinates (longitude $\lambda$, latitude $\phi'$, radius $r$) as the negative spatial gradient of a scalar potential.

$$B_m(\lambda, \phi', r, t) = -\bigtriangledown V(\lambda, \phi', r, t) \tag{1}$$

where $t$ represents the time. The potential can be expressed as a series expansion in spherical harmonics: [24].

$$V(\lambda, \phi', r, t) = a\left(\sum_{n=1}^{N}\sum_{m=1}^{n}\left(\frac{a}{r}\right)^{n+1}(g_n^m(t)\cos(m\lambda) + h_n^m(t)\sin(m\lambda)\breve{p}_m(\sin \phi')\right) \tag{2}$$

We selected N = 36 as the truncation level for the internal expansion of the World Magnetic Model. Here, 'a' (6371200 m) represents the geomagnetic reference radius, closely approximating the mean Earth radius. The variables $(\lambda, \phi', r)$ denote the longitude, latitude, and radius in a geocentric spherical reference frame, respectively. Additionally, $g_n^m(t)$ and $h_n^m(t)$ represent the time-dependent Gauss coefficients of degree n and order m, describing the Earth's main magnetic field. The parameters are defined as follows:

$$g_n^m(t) = g_n^m + \dot{g}_n^m(t - t_0) + \ddot{g}_n^m(t - t_0)^2 \tag{3}$$

$$h_n^m(t) = h_n^m + \dot{h}_n^m(t - t_0) + \ddot{h}_n^m(t - t_0)^2 \tag{4}$$

In this equation, $g_n^m, h_n^m, \dot{g}_n^m, \dot{h}_n^m, \ddot{g}_n^m, \ddot{h}_n^m$ from Eq 3 are considered constants. For any real number $\mu$, $\breve{p}_n^m(\mu)$ represents the Schmidt semi-normalized associated Legendre functions and is defined as:

$$\breve{p}_n^m(\mu) = \sqrt{2\frac{(n-m)!}{(n+m)!}}p_{n,m}(\mu) \quad, if \quad m > 0 \tag{5}$$

$$\breve{p}_n^m(\mu) = p_{n,m}(\mu) \quad, if \quad m = 0 \tag{6}$$

Finally, in the case of a data set containing hourly mean observatory data, offsets must be incorporated at each observatory to consider the local magnetic field, primarily induced in the Earth's crust, which is beyond the model's scope. Consequently, at a given observatory, the magnetic field (B) is described as follows:

$$B_m(\lambda, \phi', r, t) = -\bigtriangledown V(\lambda, \phi', r, t) + O(\lambda, \phi', r, t). \tag{7}$$

The offset vector $O(\lambda, \phi', r, t)$, commonly known as the crustal bias, remains constant over time. The parameterization is employed to model datasets selected from satellite measurements and hourly mean values observed at the observatory. The equations governing the

internal component of the field are as follows:

$$X'(\lambda, \phi', r, t) = -\frac{\partial V}{r \partial \phi'} = -\sum_{n=1}^{N} \left(\frac{a}{r}\right)^{n+2} \sum_{m=0}^{M} (g_n^m(t)\cos(m\lambda) + h_n^m(t)\sin(m\lambda)) \frac{d\breve{p}_n^m(\sin\phi')}{d\phi'}$$

$$Y'(\lambda, \phi', r, t) = -\frac{\partial V}{r\cos\phi' \partial\lambda} = \frac{1}{r\cos\phi'} \sum_{n=1}^{N} \left(\frac{a}{r}\right)^{n+2} \sum_{m=0}^{M} m(g_n^m(t)\cos(m\lambda) + h_n^m(t)\sin(m\lambda))\breve{p}_n^m(\sin\phi')$$

$$Z'(\lambda, \phi', r, t) = \frac{\partial V}{\partial r} = -\sum_{n=1}^{N}(n+1)\left(\frac{a}{r}\right)^{n+2} \sum_{m=0}^{M} (g_n^m(t)\cos(m\lambda) + h_n^m(t)\sin(m\lambda))\breve{p}_n^m(\sin\phi')$$

The equations above, with the magnetic field vector observations on the left-hand side, constitute the system of condition equations. Consequently, if there are d data points, the system comprises d linear equations involving the p parameters of the parent model:

$$y_{d\times 1} = A_{d\times p} \cdot m_{p\times 1}, \tag{8}$$

Where **y** is the column vector ($d \times 1$) of observations, **A** is the matrix ($d \times p$) of coefficients corresponding to the unknowns, which are functions of position, and **m** is the column vector ($p \times 1$) of unknowns, representing the Gauss coefficients of the model. Since there are more observations than unknowns ($d > p$), the system is over-determined and, thus, does not have an exact solution. The final main-field coefficients for 2005.0 were obtained by polynomial extrapolation of the main-field Gauss coefficients from the parent model to this date, using Eq 3. Equations, with the time-varying Gauss coefficients $g_n^m, h_n^m$ replaced by their time derivatives $\dot{g}_n^m, \dot{h}_n^m$, were then utilized to determine the final secular-variation coefficients [20].

## Haversines method for bird path change

To answer the Hypothesis stated in section, We wish to extract the directional change in the migration path of the birds from dataset 1.

A bird's decision to change its migration path is invariant intra-species but not invariant inter-species. (In other words, given a particular bird species, we may assume the path directions are independent and identically distributed (iid) directional random variables, but if species change, the assumption of iid will be violated).

## Computation of directional change in bird path on Earth surface (Column $\theta$)

We require the Haversine formula (see, for example, [25]) to compute the Column $\theta$, a Directional change in the bird's migration path.

The Haversine formula is an exact way to compute distances between two points on the surface of a sphere. This formula is essential for navigation. The formula uses the latitude and longitude of the two points. The haversine formula is a reformulation of the spherical law of cosines, but the formulation in terms of haversines is more beneficial for small angles and distances [26, 27]. The mathematical form is

$$\cos\theta = \sin\phi_A \sin\phi_B + \cos\phi_A \cos\phi_B \cos\Delta L \tag{9}$$

where $\Delta L = L_A - L_B$. Point A will have latitude $\phi_A$ and longitude $L_A$. Similarly, point B will have latitude $\phi_B$ and longitude $L_B$.

## Directional statistical preliminaries

**Von mises distribution.** A circular random variable $\theta$ follows the Von Mises distribution, also known as the Circular Normal Distribution if it is characterized by the probability density function (pdf) [28]:

$$f(\theta; \mu, k) = \frac{1}{2\pi I_0(k)} e^{k \cos \theta(\theta - \mu)}, \tag{10}$$

In this equation, $\theta$ lies in the range $[0, 2\pi)$, $\mu$ is constrained to $[0, 2\pi)$, and $(k > 0)$. The normalizing constant $I_0(k)$ is the modified Bessel function of the first kind and order zero, given by:

$$\frac{1}{2\pi} \int_1^{2\pi} \exp(k \cos \theta) d\theta = \sum_{r=0}^{\infty} \left(\frac{k}{2}\right)^{2r} \left(\frac{1}{r!}\right)^2 \tag{11}$$

To determine the cumulative distribution of the circular normal or the Von Mises Distribution, we integrate the pdf, resulting in the following cumulative distribution function (cdf):

$$F(\theta) = \frac{1}{2\pi I_0(k)} \left( \theta I_0(k) + 2 \sum_{p=1}^{\infty} \frac{I_p(k) \sin p(\theta - \mu)}{p} \right), \tag{12}$$

where $\theta$ is confined to the interval $[0, 2\pi)$.

## Circular-circular regression for overall dataset

It's a specialized statistical tool designed to analyze relationships between two sets of directional data, like the ones we used: bird migration paths and magnetic field variations (inclination and declination).

Imagine a compass where each direction (north, south, east, west) has a specific angle. Now, picture bird migration paths as lines with specific angles representing the direction the birds flew. Similarly, magnetic field variations can also be represented by angles indicating the tilt (inclination) and direction (declination) of the Earth's magnetic field at a particular location.

The circular-circular regression model essentially helps us understand how changes in these magnetic field angles (inclination and declination) might be related to the angles of bird migration paths. It takes into account the circular nature of these directional measurements, unlike traditional statistical methods that assume data points lie on a straight line.

Here are some key assumptions of the model:

- Bird migration paths and magnetic field variations can be represented by angles. This allows us to analyze them using circular statistics.

- The relationship between these angles can be described by a mathematical formula. The model helps us estimate this formula and understand how changes in magnetic field angles might influence the direction birds choose to fly.

- The errors associated with both bird path and magnetic field data are also circular. This acknowledges the inherent uncertainties in measuring directions.

By considering these assumptions, the circular-circular regression model provides a powerful tool to investigate the fascinating connection between bird navigation and the subtle variations in the Earth's magnetic field.

From the analysis done in the previous section, we find some Directional Parameters like the Bird's flight path Direction, Earth's Magnetic inclination, and declination, etc. We want to show if there is any relation between the positions of the celestial objects in our solar system affecting the direction of the bird's path. We considered the direction of the Bird's flight path as the dependent variable and other angular parameters as independent. Per our theory, we believe the Bird Path Direction $\beta$ and the other angular parameters as $\alpha$.

$(\alpha, \beta)$ have joint pdf f$(\alpha, \beta)$, $0 < \alpha, \beta \leq 2\pi$. To predict $\beta$ for a given $\alpha$, consider the regression or conditional expectation of vector $\exp(i\beta)$ given $\alpha$, say

$$E(\exp i\beta | \alpha) = \rho(\alpha)\exp(i\mu(\alpha)) = g_1(\alpha) + ig_2(\alpha) \tag{13}$$

Equivalently

$$E(\cos \beta | \alpha) = g_1(\alpha), E(\sin \beta | \alpha) = g_2(\alpha) \tag{14}$$

From which $\beta$ can be determined as

$$\mu(\alpha) = \hat{\beta} = \arctan * \frac{g_2(\alpha)}{g_1(\alpha)} \tag{18}$$

Predicting $\beta$ in this way is optional because it minimizes the below equation and is similar to the least square idea.

$$E||\exp i\beta - g(\alpha)||^2$$

Without further specification on the structure of $g_1(\alpha)$ and $g_2(\alpha)$, it isn't easy to estimate these from the data. So $g_(\alpha)$ and $g_2(\alpha)$ will be approximated by suitable functions. This leads to approximating $g_(\alpha)$ and $g_2(\alpha)$ by trigonometric polynomials of suitable degree say m

$$g_1(\alpha) \approx \sum_{k=0}^{m} (A_k \cos k\alpha + B_k \sin k\alpha), \tag{16}$$

$$g_2(\alpha) \approx \sum_{k=0}^{m} (C_k \cos k\alpha + D_k \sin k\alpha) \tag{17}$$

Thus, we have the following linear model:

$$\cos \beta = E(\cos \beta | \alpha) + \epsilon_1 = \sum_{k=0}^{m} (A_k \cos k\alpha + B_k \sin k\alpha) + \epsilon_1, \tag{18}$$

$$\sin \beta = E(\sin \beta | \alpha) + \epsilon_2 = \sum_{k=0}^{m} (C_k \cos k\alpha + D_k \sin k\alpha) + \epsilon_2 \tag{19}$$

Where $(\epsilon_1, \epsilon_2)$ is the error vector with mean vector 0 and dispersion matrix $\Sigma$ [29].

We propose a similar regression model using the response variable a change in the direction of the bird path (Variable $\theta$ calculated from our data using the Haversine formula, and independent covariate variables as Magnetic Inclination$(M_{MI})$, and Magnetic Declination$(M_{MD})$.

We assume the regression model stated in (21) to detect the significance of Magnetic field change.

$$\theta(\psi, \lambda, t) = M_{MI}(\psi, \lambda) + \rho_{MI} A_1(\psi, \lambda, t) + \epsilon_1 \tag{20}$$

$$\theta(\psi, \lambda, t) = M_{MD}(\psi, \lambda) + \rho_{MD} A_2(\psi, \lambda, t) + \epsilon_2 \tag{21}$$

The equation for all the other physicial parameters will be similar as per the Eqs (21) and (20) [29].

We assume the following components for the model.

The given components are: bird path level $\theta(\phi, \lambda, t)$ at latitude $\phi$, longitude $\lambda$ each moment (observed time) $t$ in (year/month/date) format., Magnetic Inclination($M_{MI}$), Magnetic Declination($M_{MD}$)

The unknown components are:

$\rho$ (regression coefficient signifies the directional relation between Magnetic Change and the Bird flight path direction change.), Variance of error $\epsilon$

## Watson test

[30] Provided a statistic for directional data, like the Kolmogorov-Smirnov nonparametric test statistic, to verify one sample and two sample data if the data is following uniform distribution or Von-Mises Distribution. Watson's statistic is defined by:

$$W_n^2 = \int_0^{2\pi} \left[ (F_n - F) - \int_0^{2\pi} (F_n - F) dF \right]^2 dF \tag{22}$$

If $\alpha_1, \alpha_2, \cdots, \alpha_n$ are from iid F($\alpha$) then $\alpha_{(1)} \leqslant \cdots \leqslant \alpha_{(n)}$ denote the ordered observations(with any starting point and any sense rotation) corresponding to $\alpha_1, \alpha_2, \cdots, \alpha_n$, the empirical distribution function is defined by:

Empirical distribution function:

$$F_n(\alpha) = \begin{cases} 0 & \text{for } \alpha < \alpha_{(1)} \\ \dfrac{i}{n} & \text{for } \alpha_{(i)} \leqslant \alpha \leqslant \alpha_{(i+1)} \\ 1 & \text{for } \alpha \geqslant \alpha_{(n)} \end{cases}$$

Where F = $F_0(\alpha)$

It can also be written in the form of:

$$W_n^2 = \sum_{i=1}^{n} \left[ \left( U_{(i)} - \frac{i - \frac{1}{2}}{n} \right) - \bar{U} - \frac{1}{2} \right]^2 + \frac{1}{12n}. \tag{23}$$

Here $U_i = F(\alpha_i)$. The Cramer-Von-Mises statistic can be thought of as the "second moment" of $(F_n - F)$. Watson's statistic is similar to the expression for "variance."

## Test for directional correlation

In dealing with with the circular variable $\theta$, $M_{MI}$, and $M_{MD}$ and trying to retain many of these properties,

$$\rho_{incc}(\theta_1, M_{MI1}) = \frac{E\{\sin(\theta_i - \bar{\theta})\sin(M_{MIi} - \bar{M}_{MI})\}}{\sqrt{Var(\sin(\theta_i - \bar{\theta})Var(\sin(M_{MIi} - \bar{M}_{MI}))}} \tag{24}$$

$$\rho_{decc}(\theta_1, M_{MD1}) = \frac{E\{\sin(\theta_i - \bar{\theta})\sin(M_{MDi} - \bar{M}_{MD})\}}{\sqrt{Var(\sin(\theta_i - \bar{\theta})Var(\sin(M_{MDi} - \bar{M}_{MD}))}} \tag{25}$$

## Properties of $\rho_{incc}$ and $\rho_{decc}$

1. $\rho_{incc}(\theta_1, M_{MI1})$ and $\rho_{decc}(\theta_1, M_{MD1})$ does not depend on the zero direction used for both variable

2. $|\rho_{incc}(\theta_1, M_{MI1})| \le 1$ and $|\rho_{decc}(\theta_1, M_{MD1})| \le 1$

3. $\rho_{incc}(\theta_1, M_{MI1})$ and $\rho_{decc}(\theta_1, M_{MD1}) = 0$ if $\theta$, $M_{MI}$ and $\theta$, $M_{MD}$ are independent although the converse need not to be true.

Our Hypothesis can be statistically formulated as follows:

$$H_0: \quad \rho_c = 0 \tag{26}$$

$$H_1: \quad \rho_c \neq 0. \tag{27}$$

Here we took $(\theta_1, M_{MI1}), \cdots, (\theta_n, M_{MIn})$ and $(\theta_1, M_{MD1}), \cdots, (\theta_n, M_{MDn})$ be some random sample of observations which are of two attributes both measured as angles concerning the same zero direction and the same sense of rotation. Let's take $(\theta, M_{MI})$ and $(\theta, M_{MD})$ be the joint probability density function on the torus $0 \le \theta < 2\pi$, $0 \le M_{MI} \le 2\pi$ and $0 \le M_{MD} \le 2\pi$. Let $\bar{M}_{MD}$, $\bar{M}_{MI}$ and $\bar{\theta}$ denote the mean direction of three variables.

In the Eq 28 $\bar{M}_{MD}$, $\bar{M}_{MI}$ and $\bar{\theta}$ are sample mean directions. Below correlation coefficient if $(\theta_1, M_{MI1}), \cdots, (\theta_n, M_{MIn})$ and $(\theta_1, M_{MD1}), \cdots, (\theta_n, M_{MDn})$ are random sample, given by [31]

$$r_{incc,n} = \frac{\sum_{i=1}^{n} \sin(\theta_i - \bar{\theta})\sin(M_{MIi} - \bar{M}_{MI})}{\sqrt{\sum_{i=1}^{n} \sin^2(\theta_i - \bar{\theta})\sin^2(M_{MIi} - \bar{M}_{MI})}} \tag{28}$$

$$r_{decc,n} = \frac{\sum_{i=1}^{n} \sin(\theta_i - \bar{\theta})\sin(M_{MDi} - \bar{M}_{MD})}{\sqrt{\sum_{i=1}^{n} \sin^2(\theta_i - \bar{\theta})\sin^2(M_{MDi} - \bar{M}_{MD})}} \tag{29}$$

The sample correlation is an estimation of $\rho_c$. When the joint distributions of $(\theta, M_{MI})$ and $(\theta, M_{MD})$ are not fully specified, we can use the sample measure $r_{incc,n}$ and $r_{decc,n}$ for testing the hypothesis about $\rho_c$ mentioned in (26) when $n$ is sufficiently large. For details on the test statistics and its distributional properties under $H_0$, we refer to [29].

**Table 1. Table for the watson test used for the directions of birds.**

| Bird Species | Critical Value | Test Statistics |
|---|---|---|
| Swainson's hawk | 0.061 | 0.0239 |
| Pacific loon | 0.061 | 0.0165 |
| Long-billed curlew | 0.061 | 0.0285 |
| Brown pelican | 0.061 | 0.0224 |
| Black-crowned night heron | 0.061 | 0.0488 |
| Black-Bellied Plover | 0.061 | 0.0286 |

## Results

Table 1 presents the results of the Watson test applied to the directional data representing the migration paths of various bird species. The Watson test, is a statistical test used to assess whether a set of directional data follows a von Mises distribution, which is commonly used to model circular data. Also the pictorial view of the plot is given in the Fig 2.

This section contains results in the form of Tables 2–5; results for testing the statistical hypothesis of significant directional correlation. In Tables 2 and 3, all of the $\rho$ values are given. Tables 2 and 3 show the hypothesis test result. According to our hypothesis, $\rho_c \neq 0$ happens if the value of observed $\rho$ significantly differs from zero, and p-values are significant enough, rejecting the null hypothesis. Note that bird migration depends on various factors apart from the geomagnetic change. For instance, other essential factors like food habits, Weather, Wind direction, sun and moon position, etc, are used by birds to migrate from one place to another place [2, 32]. Considering that, it makes sense to take the significance level of our hypothesis test not too high. The Discussion section (lines 235-264) highlights the novelty of your approach and the potential influence of other factors on bird migration. Could you suggest ways to incorporate or account for these additional factors in future studies? The circular correlation between the change of bird direction and the change of magnetic inclination and

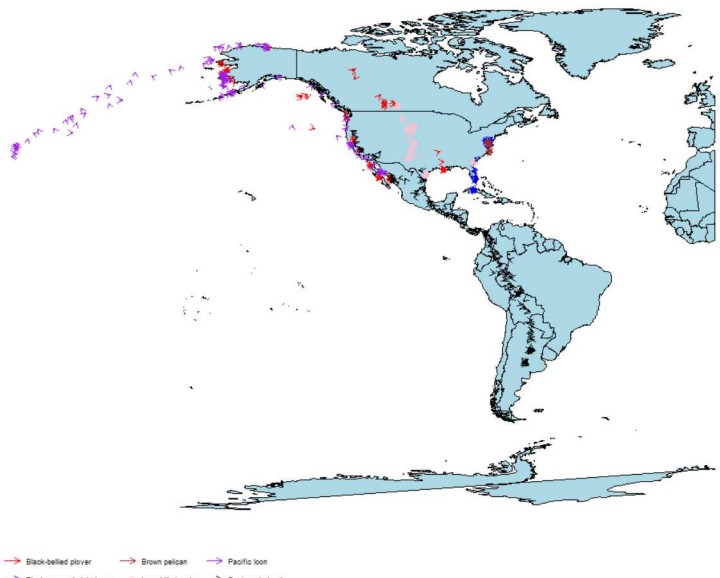

**Fig 2. Bird direction (column _θ_) plot of dataset 2.** The arrows represent the change in direction (angle) from the North-South direction. (The plots are generated by using R and ggOceanmap Package).

**Table 2. Reults of the circular-circular regression for all the birds concerning the change of the Magnetic inclination.**

| Bird Name | $\rho$ of Magnetic inclination |
| --- | --- |
| Black Crowned Night Heron | 0.2652609 |
| Black Bellies Plover | 0.06988699 |
| Brown Pelican | 0.2406245 |
| Long-billed Curlew | 0.1308153 |
| Pacific Loon | 0.1708676 |
| Swainson's Hawk | 0.1222017 |

**Table 3. Reults of the circular-circular regression for all the birds concerning the change of the Magnetic declination.**

| Bird Name | $\rho$ of Magnetic Declination |
| --- | --- |
| Black Crowned Night Heron | 0.2639779 |
| Black Bellies Plover | 0.1567983 |
| Brown Pelican | 0.1103696 |
| Long Billed Curlew | 0.08748191 |
| Pacific Loon | 0.1047719 |
| Swainson's Hawk | 0.133772 |

**Table 4. Correlation and hypothesis testing between birds' change of angle and Magnetic inclination change concerning each species.**

| Bird Name | r value | test statistics |
| --- | --- | --- |
| Pacific Loon | -0.06760874 | -0.8271246 |
| Black Bellies Plover | 0.05610714 | 0.6163309 |
| Black Crowned Night Heron | 0.2437017 | 1.763301 |
| Brown Pelican | 0.2394898 | 1.919363 |
| Long Billed Curlew | 0.01550975 | 0.1776201 |
| Swainson's Hawk | 0.05883399 | 0.7501803 |

**Table 5. Correlation and hypothesis testing between birds' change of angle and Magnetic declination change concerning each species.**

| Bird Name | r value | test statistics |
| --- | --- | --- |
| Pacific Loon | 0.119042 | 1.502856 |
| Black Bellies Plover | 0.1425152 | 1.591433 |
| Black Crowned Night Heron | -0.2358229 | -1.443047 |
| Brown Pelican | -0.01969902 | -0.1885371 |
| Long Billed Curlew | 0.05430769 | 0.6781636 |
| Swainson's Hawk | 0.09382051 | 1.241278 |

declination is given in Table 4. We find that the result of a substantial number of species supports $\rho_c \neq 0$, stating the existence of a significant directional correlation. Exceptions are only two bird species having p-values less than 0.3 and two birds with p-values less than 0.6 (Table 4); Table 5 shows similar findings.

## Discussion

1. We applied directional statistical tools to understand the migratory bird flow paths to unveil their directional dependencies on the Magnetic field change on the Earth's surface. Our initial dataset, which contains migratory bird species, latitude, longitude, and data collection date, was collected from bird-path tracking web resource [1]. Based on that, a second dataset that contains the directional change of bird path (in terms of angle) and corresponding spatiotemporal change in magnetic field change is computed. The motivation for calculating the second dataset was to frame the required dataset to test the novel hypothesis.

2. We used circular statistics to analyze the relation between the bird's direction and the magnetic field. All migration datasets and bird path directions are analyzed by setting up a field campaign to compare and validate outputs from different radar systems [33]. The datasets are circular and angular, so we will face less loss of information by using directional statistics as a tool.

3. The secondary dataset that we had to create is necessary to test our hypothesis utilizing the Haversine formula to get the direction of individual bird species using the consecutive positions of the birds.

4. We used a mathematical model for the corresponding spatio-temporal magnetic inclination and declination change.

5. We proposed a novel circular-circular regression model to investigate the hypothesis and statistically framed the hypothesis based on that regression model.

6. We assumed that each bird species subdivision's latitude and longitude distribution separately follows either circular von Mises distribution or circular uniform distribution. Such an underlying von Mises error distribution assumption is essential for circular-circular regression results. Still, it can generally be relaxed for hypothesis testing directional correlation under a large enough dataset. Testing significant directional correlation was the main objective of this paper.

7. Our result shows a statistically noteworthy non-zerodirectional association between the birds' direction change decision and the Earth's magnetic inclination and declination change.

### Broader implications

The findings of this study, particularly the support for our hypothesis based on the circular-circular regression analysis, hold significant implications for our understanding of bird navigation and have the potential for applications in conservation and migration monitoring.

**Deeper Understanding of Bird Navigation**:

- **Magnetic Field as a Navigational Cue**: This study strengthens the evidence that variations in the Earth's magnetic field (inclination and declination) play a crucial role in influencing the navigational decisions of birds during migration. It contributes to a growing body of research that sheds light on the complex sensory toolkit employed by birds for long-distance journeys.

- **Species-Specific Strategies**: The observed correlations between magnetic field variations and migratory patterns may not be universal across all bird species. Our findings pave the

way for further investigation into the diversity of navigational strategies employed by different avian groups.

- **Integration with Other Cues**: While magnetic fields appear to be an essential cue, birds likely integrate them with other environmental factors, such as wind patterns, celestial cues, and learned landmarks. Future research can explore how these various cues interact to guide bird navigation.

   **Conservation and Migration Monitoring Applications**:

- **Habitat Protection**: Understanding the influence of magnetic fields on migration routes can inform conservation efforts. By identifying crucial magnetic "waypoints" along migratory corridors, we can prioritize habitat protection and minimize human disruptions in these critical areas.

- **Improved Monitoring Tools**: The use of circular statistics and the insights gained from this study could be incorporated into the development of more sophisticated migration monitoring tools. These tools could be used to track bird movements in real-time, allowing for better prediction of migration patterns and identification of potential threats.

- **Conservation Planning**: Knowledge of magnetic field sensitivity could be factored into conservation planning strategies. For example, minimizing electromagnetic interference from human-made sources along migratory routes might be considered to avoid disruptions to avian navigation.

Overall, this study contributes valuable knowledge to the fascinating realm of bird navigation. By delving deeper into the connection between magnetic fields and migratory behavior, we can unlock insights that benefit both scientific understanding and conservation efforts. The broader implications encourage exploration of the applications of these findings in real-world scenarios, paving the way for a future where we can better protect and monitor the incredible migratory journeys of birds.

## Limitations of our study

Our study sheds light on the intriguing connection between bird migration patterns and variations in the Earth's magnetic field. However, it's crucial to acknowledge the limitations inherent to the methodologies employed, which call for cautious interpretation of the results and pave the way for future research directions.

**Bird path tracking data.** We relied on a single web resource for bird path tracking data. This introduces potential biases. The dataset might not encompass the complete diversity of bird species or migratory behaviors. Additionally, the accuracy of the data depends on the tracking methods employed (e.g., GPS tags) and potential limitations associated with those methods (e.g., tag attachment affecting bird behavior, data gaps due to signal loss).

**Magnetic field data.** The World Magnetic Model (WMM) offers a coarse representation of the Earth's magnetic field. As discussed earlier, its static nature and limited resolution might not capture the dynamic variations birds might be sensitive to. Ideally, real-time magnetic field data, if available with sufficient coverage for migratory routes, could provide a more accurate picture.

## Analytical method limitations

**Circular statistics assumption.** The study utilizes circular statistics assuming a von Mises distribution for bird directions. While this approach is well-suited for analyzing directional

data, it might not fully capture the complexities of bird navigation strategies. Some species might exhibit more nuanced or variable flight patterns that deviate from a single underlying distribution.

**Circular-circular regression model complexity.** The circular-circular regression model employed is a powerful tool, but its complexity can make it challenging to interpret the specific mechanisms behind the observed correlations between magnetic field variations and bird navigation. Further research could explore complementary methods to elucidate the underlying biological processes.

## Scope limitations

**Species specificity.** The study investigates a limited set of bird species. The observed correlations between magnetic field variations and migratory patterns might not be universal across all avian species. Further studies exploring a broader range of birds are necessary to determine the generalizability of these findings.

**Focus on magnetic field.** The study focuses solely on the influence of magnetic field variations. Other environmental factors, such as wind patterns, food availability, and celestial cues, likely play a role in bird navigation. A more holistic understanding requires incorporating these additional factors into future studies.

By acknowledging these limitations, we can refine future research efforts in this field. Employing a wider variety of data sources, incorporating real-time environmental measurements, and exploring more nuanced analytical techniques will enable us to gain a deeper understanding of the remarkable navigational abilities of birds.

## Conclusions

We statistically examined the potential relationship between the directional change in Magnetic inclination and declination at a given location and a bird's decision to alter its path. We proposed a novel circular-circular regression model with von Mises error assumption to test our hypothesis. Most results from the hypothesis test using circular-circular regression analysis supported the hypothesis that the change in birds' migration paths depends on the corresponding change in Magnetic inclination and declination. These findings suggest that "magnetic murmurs" may play a role in avian navigation; immediate future work could be to improve the underlying model.

## Future work

Building upon the insights gained from this study, future research can delve deeper into the fascinating world of avian navigation and its relationship with environmental factors, particularly the influence of magnetic inclination and declination on migratory patterns. Here are some exciting avenues to explore:

### Incorporating a Bayesian directional model

The current circular-circular regression model provides valuable insights, but it might not capture the full complexity of bird navigation. Future studies could explore using a more sophisticated framework, such as a hierarchical Bayesian model with a directional component. This would allow for incorporating prior information about bird behavior and environmental factors, leading to potentially more nuanced and robust results. Additionally, the Bayesian framework would facilitate the integration of uncertainties associated with data sources and model parameters.

### Accounting for spatial and temporal variations

While the current study investigated the influence of magnetic fields on migration patterns, these fields exhibit spatial and temporal variations. Future research could leverage spatiotemporal statistical models to account for these variations and gain a more comprehensive understanding of how birds navigate across different geographic regions and throughout their migratory journeys. This could involve incorporating high-resolution magnetic field data alongside bird tracking information.

### Investigating multi-sensory integration

Birds likely utilize a complex interplay of senses for navigation, not just relying solely on magnetic fields. Future studies could explore the integration of data on other potentially influential factors, such as wind patterns, celestial cues (star positions, sun compass), and olfactory cues (smells) using advanced statistical techniques. This multi-sensory approach could provide a more holistic understanding of how birds navigate their environment.

### Development of species-specific models

The current study explored a limited set of bird species. Future research could develop species-specific models by incorporating biological and ecological information about different birds. This would allow for a more tailored understanding of the unique navigational strategies employed by various avian species.

### Integration with machine learning techniques

Machine learning algorithms hold promise for uncovering complex relationships between environmental factors and bird migration patterns. Future research could explore the integration of machine learning tools with the statistical models developed in this study. This could lead to the identification of previously unknown patterns or interactions that influence bird navigation.

Overall, these proposed future research directions aim to further our understanding of the intricate relationship between avian navigation and the complex interplay of environmental factors. By building upon the foundation laid by this study and exploring these innovative approaches, we can unlock the secrets that guide these feathered voyagers on their remarkable migratory journeys.

## Acknowledgments

Prithwish Ghosh is grateful to North Carolina State University, Dr. Debashis Chatterjee is grateful to Visva Bharati, Santiniketan, Dr. Amlan Banerjee, and Dr. Shiladri Shekhar Das are thankful to the Indian Statistical Institute, Kolkata, for the facilities provided to the researchers. Dr. Debashis Chatterjee thanks the late Prof. Sarbani Patranabis Deb (Geological Studies Unit, Indian Statistical Institute, Kolkata) for her insightful remarks regarding this research.

## Author Contributions

**Conceptualization:** Prithwish Ghosh, Debashis Chatterjee.

**Data curation:** Prithwish Ghosh.

**Formal analysis:** Prithwish Ghosh.

**Investigation:** Prithwish Ghosh, Debashis Chatterjee.

**Methodology:** Prithwish Ghosh.

**Project administration:** Prithwish Ghosh, Debashis Chatterjee, Amlan Banerjee, Shiladri Shekhar Das.

**Resources:** Prithwish Ghosh.

**Software:** Prithwish Ghosh.

**Supervision:** Debashis Chatterjee, Amlan Banerjee, Shiladri Shekhar Das.

**Validation:** Debashis Chatterjee, Amlan Banerjee, Shiladri Shekhar Das.

**Writing – original draft:** Prithwish Ghosh.

**Writing – review & editing:** Debashis Chatterjee.

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
