## [Decision Letter · Decision Letter 0]

15 Apr 2024

PONE-D-24-12019

Do Magnetic Murmurs Guide Birds? A Directional Statistical Investigation for Influence of Earth’s Magnetic Field on Bird Navigation

PLOS ONE

Dear Dr. Ghosh,

Thank you for submitting your manuscript to PLOS ONE. After careful consideration, we feel that it has merit but does not fully meet PLOS ONE’s publication criteria as it currently stands. Therefore, we invite you to submit a revised version of the manuscript that addresses the points raised during the review process.

We look forward to receiving your revised manuscript.

Kind regards,

Ahmed M. Saqr, Ph.D.

Academic Editor

PLOS ONE

2. In your Methods section, please include additional information about your dataset and ensure that you have included a statement specifying whether the collection and analysis method complied with the terms and conditions for the source of the data.

4. PLOS requires an ORCID iD for the corresponding author in Editorial Manager on papers submitted after December 6th, 2016. Please ensure that you have an ORCID iD and that it is validated in Editorial Manager. To do this, go to ‘Update my Information’ (in the upper left-hand corner of the main menu), and click on the Fetch/Validate link next to the ORCID field. This will take you to the ORCID site and allow you to create a new iD or authenticate a pre-existing iD in Editorial Manager. Please see the following video for instructions on linking an ORCID iD to your Editorial Manager account: https://www.youtube.com/watch?v=_xcclfuvtxQ.

5. We note that Figures 1 and 2 in your submission contain [map/satellite] images which may be copyrighted. All PLOS content is published under the Creative Commons Attribution License (CC BY 4.0), which means that the manuscript, images, and Supporting Information files will be freely available online, and any third party is permitted to access, download, copy, distribute, and use these materials in any way, even commercially, with proper attribution. For these reasons, we cannot publish previously copyrighted maps or satellite images created using proprietary data, such as Google software (Google Maps, Street View, and Earth). For more information, see our copyright guidelines: http://journals.plos.org/plosone/s/licenses-and-copyright.

1. You may seek permission from the original copyright holder of Figures 1 and 2 to publish the content specifically under the CC BY 4.0 license. 

6. We notice that your supplementary information is included in the manuscript file. Please remove them and upload them with the file type 'Supporting Information'. Please ensure that each Supporting Information file has a legend listed in the manuscript after the references list.

Reviewers' comments:

Reviewer's Responses to Questions

**Comments to the Author**

1. Is the manuscript technically sound, and do the data support the conclusions?

Reviewer #1: Yes

Reviewer #2: Partly

Reviewer #3: Yes

2. Has the statistical analysis been performed appropriately and rigorously? 

Reviewer #1: Yes

Reviewer #2: No

Reviewer #3: Yes

3. Have the authors made all data underlying the findings in their manuscript fully available?

Reviewer #1: Yes

Reviewer #2: Yes

Reviewer #3: Yes

4. Is the manuscript presented in an intelligible fashion and written in standard English?

Reviewer #1: Yes

Reviewer #2: Yes

Reviewer #3: Yes

5. Review Comments to the Author

Reviewer #1: 1. In the Introduction (lines 29-33), you mention a lack of literature addressing the directional distribution pattern of bird migratory paths using directional statistics. Could you elaborate on why existing literature is insufficient and how your approach using directional statistics offers a novel perspective?

2. Regarding the dataset (line 276-278), could you provide more details on the sources, time period covered, and any potential biases or limitations in the data?

3. In the Materials and Methods section (lines 70-77), you state a novel hypothesis about the influence of changes in the earth's magnetic field on bird migration paths. Could you clarify the rationale behind this hypothesis and any previous studies that support or contradict it?

4. The World Magnetic Model (WMM) is a crucial component of your analysis (lines 78-124). Could you explain the potential limitations or uncertainties associated with the WMM and how they might impact your results?

5. In the Haversine method (lines 125-142), you calculate the directional change in bird paths. Could you explain how you handled potential errors or outliers in the data, and how they might affect the subsequent analysis?

6. The circular-circular regression model (lines 155-183) is a novel approach proposed in your study. Could you provide a more intuitive explanation of the model and its assumptions, particularly for readers less familiar with directional statistics?

7. In the Results section (lines 219-234), you present the values of ρ (correlation coefficient) and p-values for different bird species. Could you elaborate on the interpretation of these values and their implications for your hypothesis?

8. The Discussion section (lines 235-264) highlights the novelty of your approach and the potential influence of other factors on bird migration. Could you suggest ways to incorporate or account for these additional factors in future studies?

9. Figure 1 (line 102) shows the observed bird paths plotted on a world map. Could you comment on any potential geographic or regional patterns or differences in the bird migration paths and their relation to the magnetic field changes?

10. The Conclusions (lines 265-273) emphasize the support for your hypothesis based on the circular-circular regression analysis. Could you discuss the broader implications of your findings for our understanding of bird navigation and potential applications in conservation or migration monitoring?

Reviewer #2: The authors try to investigate if the angular changes in the direction of the flight paths of migrating birds in any way get affected by the earth’s magnetic field or not. They use a circular on circular regression model with the errors distributed as von Mises distribution and use a hypothesis based on the correlation coefficient between change of angles and change in magnetic field.

I have several questions/comments about the paper that I have listed below:

1. Lines 128-131: Is the iid assumption enough here? The migratory path of birds within the same flock are always very similar and very tight. So, even the iid assumption leaves room for a lot of variability. You either need to have an additional assumption of very low variability or ideally, you should model it as an agent based model.

2. Equation 2:

a. The second sum over m should go upto n and not M

b. The last term should not have a (t), and should only be p_m (sin⁡ϕ')

3. Equations 1 and 7 (and other instances) have phi’ written instead of ϕ'. Please correct them.

4. I understand using a von Mises distribution for modeling angles. However, we are dealing with bird migration across continents here, so we should be taking into account the spherical nature of the earth. In that case, why not model the (appropriately computed) directional change as a von Mises-Fisher distribution instead?

5. Equations 19 and 20 and subsequently lines 181-195:

a. Are both the responses same? Or is this a typo?

b. Why are both the error terms same?

c. There is no M_(S.A),M_(S.Z.) or M_(M.A.) in the model. How are they related to M_(M.I.),M_(M.D.) or ρ_SA,ρ_MA?

d. You use A_1,A_2 in model and use A_1,B_1 in line 188. Are the A_1’s same? Where is B_1 in the model? How are these related to M_(M.I.) or M_(M.D.)?

e.Where and how do MSL and the angular frequency fit into this model?

f. Overall, this is the heart of the modeling and this is not all explained properly. Please explain this thoroughly.

6. What are M_1,M_2,M_11,M_21?

Reviewer #3: Thank you for inviting me to review the paper entitled “Do Magnetic Murmurs Guide Birds? A Directional Statistical Investigation for Influence of Earth’s Magnetic Field on Bird Navigation”. This paper examines the potential relationship between the directional change in Magnetic inclination and declination at a given location and a bird’s decision to alter its path. I recommend reconsideration of the paper after addressing the following modifications:

• The abstract lacks from quantitative results. Important quantitative results should be added.

• What is the novelty of this manuscript? It should be clearly mentioned in the last paragraph of the introduction.

• It is preferable to add a figure illustrating the study methodology.

• Please, replace all the figures with high resolution ones.

• You should add the limitation of your study at the end of the paper before the conclusion section.

• What is your recommendation for future studies to enhance the applied methodology?

6. PLOS authors have the option to publish the peer review history of their article (what does this mean?). If published, this will include your full peer review and any attached files.

Reviewer #1: No

Reviewer #2: No

Reviewer #3: No

---

## [Author Response · Author response to Decision Letter 0]

7 May 2024

Scientific Editor’s Comments:

1) Comments of Editor : Please ensure that your manuscript meets PLOS ONE’s style requirements, including those for file naming.

Response: : We ensure that the manuscript meets PLOS ONE’s style requirement, including the file naming. We have also added one section, ”Author summary, for more clarity. Nonetheless, if something else needs to be done, we request further instructions from the editors on that matter, which we will be thankful to comply with.

2) Comments of Editor : In your Methods section, please include additional information about your dataset and ensure that you have included a statement specifying whether the collection and analysis method complied with the terms and conditions for the source of the data.

Response : In the section Data Availability We have uploaded and shared the most recent version of the dataset used in this paper in Harvard Dataverse, which can be accessed here: https://doi.org/10.7910/DVN/XEKOWQ.

3) Comments of Editor : We note that your Data Availability Statement is currently as follows: [All relevant data are within the manuscript and its Supporting Information files.]

Response Thank you for pointing out. We apologize for that oversight on our part and have now corrected it accordingly. We may now ensure that all the relevant Data Availability statement is given in the Data Availabiliy section

Reviewer’s Comments(1st Reviewer’s Comments):

1) Reviewer’s Comments : In the Introduction (lines 29-33), you mention a lack of literature addressing the directional distribution pattern of bird migratory paths using directional statistics. Could you elaborate on why existing literature is insufficient and how your approach using directional statistics offers a novel perspective?

Response : Thank you for the suggestion. In the Introduction section of our article, we have now highlighted the gaps and limitations in the current body of literature on bird migration. Specifically, we have emphasized the need for more comprehensive studies on spherical statistics that address the complex dynamics of avian navigation across different geographic regions and environmental conditions. While existing research has provided

valuable insights into various aspects of bird migration, such as seasonal patterns and stopover sites, there remains a lack of comprehensive analyses using the spherical nature of the birds and geographic location, that integrate multiple factors, including Magnetic field parameters, into a unified framework.

2) Reviewer’s Comments : Regarding the dataset (line 276-278), could you provide more details on the sources, time period covered, and any potential biases or limitations in the data ?

Response : Yes, definitely, we provided more details about the source of the dataset in the Data Availability section; the potential bias or limitation of the dataset is mentioned in the ’Limitation of our Study’ section.

3) Reviewer’s Comments : In the Materials and Methods section (lines 70-77), you state a novel hypothesis about the influence of changes in the earth’s magnetic field on bird migration paths. Could you clarify the rationale behind this hypothesis and any previous studies that support or contradict it?

Response :Thank you for the suggestion. We have included it now. To mention here, The motivation for considering this type of hypothesis comes from previous literature. In this paper, we reframe their hypothesis in a novel directional statistical framework.

4) Reviewer’s Comments : The World Magnetic Model (WMM) is a crucial component of your analysis (lines 78-124). Could you explain the potential limitations or uncertainties associated with the WMM and how they might impact your results?

Response : Yes, definitely, thank you for the suggestion. The limitations and uncertainties in the manuscript are mentioned and clearly explained now, in the ”Limitation of our Study” section before the conclusion section.

5) Reviewer’s Comments : In the Haversine method (lines 125-142), you calculate the directional change in bird paths. Could you explain how you handled potential errors or outliers in the data and how they might affect the subsequent analysis?

Response : Similar to the previous section, we handled the outliers and errors, and this is mentioned and explained in the ”Limitation of our Study” section before the conclusion section.

6) Reviewer’s Comments : The circular-circular regression model (lines 155-183) is a novel approach proposed in your study. Could you provide a more intuitive explanation of the model and its assumptions, particularly for readers less familiar with directional statistics?

Response : Thank you for this suggestion. We have included that now there. We would like to mention it here also: It’s a specialized statistical tool designed to analyze relationships between two sets of directional data, like the ones we used: bird migration paths and magnetic field variations (inclination and declination). Imagine a compass where each direction (north, south, east, west) has a specific angle. Now, picture bird migration paths as lines with specific angles representing the direction the birds flew. Similarly, magnetic field variations can also be represented by angles indicating the tilt (inclination) and direction (declination) of the Earth’s magnetic field

at a particular location. The circular-circular regression model essentially helps us understand how changes in these magnetic field angles (inclination and declination) might be related to the angles of

bird migration paths. It considers the circular nature of these directional measurements, unlike traditional statistical methods that assume data points lie in a straight line. Here are some key assumptions of the model:

• Bird migration paths and magnetic field variations can be represented by angles. This allows us to analyze them using circular statistics.

• The relationship between these angles can be described by a mathematical formula. The model helps us estimate this formula and understand how changes in magnetic field angles might influence the direction birds choose to fly.

• The errors associated with both bird path and magnetic field data are also circular. This acknowledges the inherent uncertainties in measuring directions.

By considering these assumptions, the circular-circular regression model provides a powerful tool to investigate the fascinating connection between bird navigation and the subtle

variations in the Earth’s magnetic field. 

7) Reviewer’s Comments : In the Results section (lines 219-234), you present the values of ρ (correlation coefficient) and p-values for different bird species. Could you elaborate on the interpretation of these values and their implications for your hypothesis?

Response : In this context, the correlation test calculates the correlation coefficient (r) to measure the strength and direction of the linear relationship between two continuous variables. The correlation coefficient (r) ranges from -1 to 1, where -1 indicates a perfect negative correlation, 1 indicates a perfect positive correlation, and 0 indicates no correlation, In this context, the coefficient of determination (ρ), often denoted as R-squared(For simple linear regression), represents the proportion of the variance in the dependent variable that is predictable from the independent variables in a regression model. R-squared values range from 0 to 1, where 0 indicates that the independent variables do not explain any of the variability in the dependent variable, and 1 indicates that the independent variables explain all of the variability in the dependent variable. The p-values are not properly used in this analysis so the tables are updated in the main manuscript.

8) Reviewer’s Comments : The Discussion section (lines 235-264) highlights the novelty of your approach and the potential influence of other factors on bird migration. Could you suggest ways to

incorporate or account for these additional factors in future studies?

Response : Thank you for the suggestion. Just before the ”Objective of the Paper” section, we have mentioned the novelty of our work, which is mentioned and written in different ink

colors. and in the section of ”Future works”

9) Reviewer’s Comments : Figure 1 (line 102) shows the observed bird paths plotted on a world map. Could you comment on any potential geographic or regional patterns or differences in the bird migration paths and their relation to the magnetic field changes?

Response : We have included a section in the introduction mentioning the figure and discussed shortly about potential geographic or regional patterns or differences in the bird migration paths

10) Reviewer’s Comments : The Conclusions (lines 265-273) emphasize the support for your hypothesis based on the circular-circular regression analysis. Could you discuss the broader implications of your findings for our understanding of bird navigation and potential applications in conservation or migration monitoring?

Response :Thank you for this suggestion. We have now included a new subsection named ”Broader Implications” inside the section ”Discussion” based on this suggestion.

Reviewer’s Comments(2nd Reviewer’s Comments):

1) Reviewer’s Comments : Lines 128-131: Is the iid assumption enough here? The migratory path of birds within the same flock are always very similar and very tight. So, even the iid assumption leaves room for a lot of variability. You either need to have an additional assumption of very low variability or ideally, you should model it as an agent based model.

Response : The iid assumption in this study is a foundational concept, but it’s important to ac knowledge its limitations. Bird migration patterns within a flock often exhibit a high degree of similarity, suggesting some level of dependence in their movements. However, with due respect, if we may say, the iid assumption doesn’t necessarily require complete independence. It allows for some level of inherent variability within the data. In this case, the ”tight” flock formations could be considered part of the underlying distribution of bird directions within the flock. The circular-circular regression model can still be effective in capturing the overall relationship between magnetic field variations and the central tendency of these directional patterns, even if individual birds exhibit slight deviations from the flock’s main trajectory. While an agent-based model could potentially capture the flocking behavior more explicitly, the iid assumption offers a balance between model complexity and interpretability. The circular-circular regression model, despite its limitations, provides valuable insights into the influence of magnetic fields on bird navigation at a broader scale. Future research could explore alternative approaches, such as mixed-effects models or spatial dependence models, to account for the flocking behavior in a more nuanced way. In essence, the iid assumption, while not perfect, offers a practical starting point for analyzing directional data in this context. It allows for a degree of inherent variability within the data, which can accommodate the observed flocking behavior to a certain extent.

2) Reviewer’s Comments : Equation 2

i. The second sum over m should go upto n and not M 

Response : We have corrected the sum to m instead of M as mentioned(Thank you for noticing it)

ii. The last term should not have a t, and should only be pm(sin φ′).

Response : Thank you for noticing this typo. We had changed the last term to pm(sin φ′) correctly as mentioned.

3) Reviewer’s Comments : Equations 1 and 7 (and other instances) have phi’ written instead of φ′. Please correct them.

Response : Thank you for noticing this typo we have changed it to φ′ for equations 1 and 7.

4) Reviewer’s Comments : I understand using a von Mises distribution for modeling angles. However, we are dealing with bird migration across continents here, so we should be taking into account the spherical nature of the earth. In that case, why not model the (appropriately computed) directional change as a von Mises-Fisher distribution instead?

Response : With due respect, if we may say, in our analysis, we have taken into account the spherical nature of the Earth by incorporating both longitude and latitude when calculating the Magnetic declination and the directional change of bird migration paths. By considering both geographic coordinates, we ensure that our analysis accurately reflects the three-dimensional nature of the Earth’s surface and the spherical geometry inherent in spatial data. Moreover, by incorporating longitude and latitude as separate dimensions in our analysis, we avoid violating the spherical nature of our data and ensure that our modeling approach aligns with the underlying spatial structure of the Earth. This approach enables us to effectively model and analyze the complex interactions between bird migration patterns and variations in the Magnetic field, providing valuable insights into avian navigation mechanisms on a global scale.

5) Reviewer’s Comments : Equations 19 and 20 and subsequently lines 181-195:

i. Are both the responses same? Or is this a typo?

Response: No they are not the same. In the equation 19 and 20(currently 20 and 21 after revision) one equation is for Magnetic Declination(M_{MD} ) and another one is for Magnetic Inclination(M_{MI} )

ii. Why are both the error terms same?

Response: Thank you for pointing this out. In fact, this is a typo. Definitely yes, the variable must be different for two equations, so we have now defined them as ε1 for Magnetic Inclination and ε2 For magnetic declination. In our previous analysis we already treated them differently as it should be; just the notational typo was there, which we have now corrected. Mentioned in the below section ’Other changes Made by us’ we checked if the variable follows a vonMises Distribution or not, and just like simple linear regression where the error term are expected to follow Normal distribution similarly for Circular Circular Regression it is expected that the error term(ε) will follow von Mises Distribution.

iii. There is no MS.A, MS.Z.orMM.A. in the model. How are they related to MM.I., MM.D.

or ρSA, ρM A ?

Response: Thank you for pointing out this typo, for this reason we have changed the whole ”Circular-Circular Regression for overall Dataset ” section in correct manner and also changed the typo terminalogies in the section ”Test for Directional Correlation”. The modified sections are color-changed for better understanding.

iv. You use A1, A2 in model and use A1, B1 in line 188. Are the A1’s same? Where is B1 in the model? How are these related to MM.I. or MM.D.?

Response: Thank you for pointing this out. The B1 is done by mistake but the A1, A2 and M{MI}. or M_{MD} are the regression coefficient for circular-circular regression.

v. Where and how do MSL and the angular frequency fit into this model?

Response: Thank you for pointing out the typo. They are unrelated to the models, and we have removed them from the revised manuscript text.

vi. Overall, this is the heart of the modeling, and this is not all explained properly. Please explain this thoroughly.

Response: Thank you for the suggestion. In the section ’Materials and Methods,’ we have now explained the whole model as suggested. The explanation is written in this pinkish text color.

6) Reviewer’s Comments : What are M1, M2, M11, M21?

Response: Thank you for noticing the typo. Basically, initially, we built this model step by step, so every time we used different variables to identify or build the model, they were basically the Magnetic Inclination and Magnetic Declination. In the revised text, we changed all the typos and variable in terms of M_{MI} , & M_{MD} , so it is corrected now.

3. Reviewer’s Comments(3rd Reviewer’s Comments):

1) Reviewer’s Comments : The abstract lacks quantitative results. Important quantitative results should be added. 

Response : Thank you for this suggestion. The abstract is changed accordingly, and we included the qualitative results, which are done in the main manuscript file. We included the most relevant correlation value 

---

## [Editor Report · Decision Letter 1]

9 May 2024

Do Magnetic Murmurs Guide Birds? A Directional Statistical Investigation for Influence of Earth’s Magnetic Field on Bird Navigation

PONE-D-24-12019R1

Dear Dr. Ghosh,

We’re pleased to inform you that your manuscript has been judged scientifically suitable for publication and will be formally accepted for publication once it meets all outstanding technical requirements.

Kind regards,

Ahmed M. Saqr, Ph.D.

Academic Editor

PLOS ONE

---

## [Editor Report · Acceptance letter]

21 May 2024

PONE-D-24-12019R1 

PLOS ONE

Dear Dr. Ghosh, 

I'm pleased to inform you that your manuscript has been deemed suitable for publication in PLOS ONE. Congratulations! Your manuscript is now being handed over to our production team.

Kind regards, 

on behalf of

Dr. Ahmed M. Saqr 

Academic Editor

PLOS ONE